FunPred 3.0: improved protein function prediction using protein interaction network

http://orcid.org/0000-0001-9251-8656 Saha Sovan 1
Chatterjee Piyali 2
http://orcid.org/0000-0003-1780-0461 Basu Subhadip 3
Nasipuri Mita 3
http://orcid.org/0000-0002-3840-7610 Plewczynski Dariusz 4 5 d.plewczynski@cent.uw.edu.pl
1 Department of Computer Science and Engineering, Dr. Sudhir Chandra Sur Degree Engineering College , Kolkata, West Bengal , India
2 Department of Computer Science and Engineering, Netaji Subhash Engineering College , Kolkata , India
3 Department of Computer Science and Engineering, Jadavpur University , Kolkata, West Bengal , India
4 Laboratory of Functional and Structural Genomics, Centre of New Technologies, University of Warsaw , Warsaw , Poland
5 Faculty of Mathematics and Information Science, Warsaw University of Technology , Warsaw , Poland
Orlov Yuriy
Electronic publication date: 2019 May 22
Publication date: 2019
Volume: 7
Electronic Location ID: e6830
Received 2018 Jul 20; Accepted 2019 Mar 21
Copyright: © 2019 Saha et al.
Copyright year: 2019
Copyright holder: Saha et al.
License: This is an open access article distributed under the terms of the Creative Commons Attribution License, which permits unrestricted use, distribution, reproduction and adaptation in any medium and for any purpose provided that it is properly attributed. For attribution, the original author(s), title, publication source (PeerJ) and either DOI or URL of the article must be cited.
License URL: https://creativecommons.org/licenses/by/4.0/

Keywords: Protein–protein interactions, Protein interaction networks, Neighborhood approach, MIPS Database, Protein function prediction, Physico-chemical properties

Funding: Center for Microprocessor Applications for Training Education and Research University Grants Commission Research Award from University Grants Commission, Government of India F.30-31/2016(SA-II) Department of Biotechnology BT/PR16356/BID/7/596/2016 Ministry of Science and Technology, Government of India Polish National Science Centre 2014/15/B/ST6/05082 Foundation for Polish Science (TEAM to Dariusz Plewczynski) and the grant from Department of Science and Technology, Govt. of India and Polish Government under Indo-Polish/Polish-Indo DST/INT/POL/P-36/2016 Nucleome Positioning System for Spatiotemporal Genome Organization and Regulation” within 4DNucleome National Institute of Health program 1U54DK107967-01 European Commission as European Cooperation in Science and Technology European Commission as European Cooperation in Science and Technology actions CA18127 “International Nucleome Consortium” (INC), and CA16212 “Impact of Nuclear Domains On Gene Expression and Plant Traits” RENOIR Project by the European Union Horizon 2020 research and innovation programme under the Marie Skłodowska-Curie grant agreement No 691152 and by Ministry of Science and Higher Education (Poland) W34/H2020/2016, 329025/PnH/2016 The authors received support (infrastructure facilities) from the “Center for Microprocessor Applications for Training Education and Research” research laboratory of the Computer Science Department, Jadavpur University, India. This project is partially supported by a University Grants Commission Research Award (F.30-31/2016(SA-II)) from the University Grants Commission, Government of India, and the Department of Biotechnology project (No. BT/PR16356/BID/7/596/2016), Ministry of Science and Technology, Government of India. This work has been co-supported by the Polish National Science Centre (2014/15/B/ST6/05082), the Foundation for Polish Science (TEAM to Dariusz Plewczynski) and the grant from Department of Science and Technology, Govt. of India and Polish Government under Indo-Polish/Polish-Indo project No.: DST/INT/POL/P-36/2016. The work was co-supported by grant 1U54DK107967-01 “Nucleome Positioning System for Spatiotemporal Genome Organization and Regulation” within the 4DNucleome National Institute of Health program, and by the European Commission as European Cooperation in Science and Technology European Commission as European Cooperation in Science and Technology actions: CA18127 “International Nucleome Consortium” (INC), and CA16212 “Impact of Nuclear Domains On Gene Expression and Plant Traits.” The work was partially supported as RENOIR Project by the European Union Horizon 2020 research and innovation programme under the Marie Skłodowska-Curie grant agreement No 691152 and by the Ministry of Science and Higher Education (Poland), grant numbers W34/H2020/2016, 329025/PnH/2016. The funders had no role in study design, data collection and analysis, decision to publish, or preparation of the manuscript.

==============================
Proteins are the most versatile macromolecules in living systems and perform crucial biological functions. In the advent of the post-genomic era, the next generation sequencing is done routinely at the population scale for a variety of species. The challenging problem is to massively determine the functions of proteins that are yet not characterized by detailed experimental studies. Identification of protein functions experimentally is a laborious and time-consuming task involving many resources. We therefore propose the automated protein function prediction methodology using in silico algorithms trained on carefully curated experimental datasets. We present the improved protein function prediction tool FunPred 3.0, an extended version of our previous methodology FunPred 2, which exploits neighborhood properties in protein–protein interaction network (PPIN) and physicochemical properties of amino acids. Our method is validated using the available functional annotations in the PPIN network of Saccharomyces cerevisiae in the latest Munich information center for protein (MIPS) dataset. The PPIN data of S. cerevisiae in MIPS dataset includes 4,554 unique proteins in 13,528 protein–protein interactions after the elimination of the self-replicating and the self-interacting protein pairs. Using the developed FunPred 3.0 tool, we are able to achieve the mean precision, the recall and the F-score values of 0.55, 0.82 and 0.66, respectively. FunPred 3.0 is then used to predict the functions of unpredicted protein pairs (incomplete and missing functional annotations) in MIPS dataset of S. cerevisiae. The method is also capable of predicting the subcellular localization of proteins along with its corresponding functions. The code and the complete prediction results are available freely at: https://github.com/SovanSaha/FunPred-3.0.git.

Introduction

Proteins with similar functions are more likely to interact. If the function of one protein is known then the functions of the binding un-annotated protein may either be experimentally assigned or computationally predicted (Chatterjee et al., 2011a, 2011b; Moosavi, Rahgozar & Rahimi, 2013; Prasad et al., 2017; Saha et al., 2012, 2014; Sriwastava, Basu & Maulik, 2015). Several computational techniques have been developed using either the protein sequence (Ng & Henikoff, 2003), protein structure (Lee, Redfern & Orengo, 2007; Mills, Beuning & Ondrechen, 2015), protein–protein interactions (Moosavi, Rahgozar & Rahimi, 2013; Schwikowski, Uetz & Fields, 2000; Vazquez et al., 2003; Xiong et al., 2013), or sequence motifs or signatures (Chatterjee et al., 2011a; Chen et al., 2007; Lichtarge, Bourne & Cohen, 1996). Protein interaction datasets are represented as graphs (with every node corresponding to an individual protein and each edge between a pair of nodes representing the interaction between them) can be used to assign biological functions to a protein with an assumption that close neighbors of a protein are functionally similar.

The protein function prediction problem is characterized by several factors like the diversity of members for functional groups, the hierarchical relationships among functional classes, incomplete or missing information about proteins and their functions. Thus, it defines a complex multi-label learning problem (Jiang & McQuay, 2012; Valentini, 2014; Zhang & Zhou, 2014). Hierarchical relationships among labels are described in Munich information center for protein (MIPS) functional catalogue and gene ontology. Valentini (2014) uses a binary classifier for each label according to true path rule and the funCat. Recent work of Guoxian and the co-authors (Yu, Zhu & Domeniconi, 2015), explored the incomplete label problem in a hierarchical manner using function correlation. Another approach for predicting protein function, as proposed by Piovesan et al. (2015), includes the combination of the trio: PPIN information, protein domain and sequence. In another work, Zhao et al. (2016) invokes dynamic weighted interaction network instead of the static one. This dynamic network is enriched with PPIN, time course gene expression data, protein’s domain information and protein complex information which ultimately predict function of a protein using majority ranking. While most of the predictive models highlights on the most highly related similar proteins in the neighborhood of the test protein, Reinders, Van Ham & Makrodimitris (2018) focuses on the less similar proteins. It is shown by the application of label-space dimensionality reduction techniques that though these proteins are less similar but they are quite informative and plays an important role in protein function prediction. Another iterative algorithm is implemented by Sun et al. (2018) for predicting protein functions. It is completely dependent on the identification of the functional dependencies which are based on proteins and their interactions. The sequence similarity network is another important aspect for protein function prediction, which is considered in the development of the Effusion methodology as proposed in the work of Yunes & Babbitt (2018). Other notable works in this field are Wang et al. (2018) and Fa et al. (2018).

All these methods discussed above have already taken protein function prediction to the next higher level. Yet, the still-uncovered details in the study and analysis support the need for new computational methods exploiting the protein–protein interaction networks for biological function identification. In our novel methodology FunPred 3.0, the functions of test proteins are determined by analyzing the neighborhood properties of their protein interaction network. At the same time, certain selected physicochemical properties of amino acids are also used along with it. This task is challenging because of several reasons; for example, large number of functional groups, different levels of the interconnection hierarchy, proteins with multiple functional groups, and incomplete or missing labels. In this proposed methodology, the MIPS dataset (Mewes et al., 2002) is used. It contains protein pairs along with their corresponding functions. At the initial phase, to estimate the effectiveness of FunPred 3.0, essential proteins are selected as test proteins. The functions of these proteins are considered to be unknown for experimental purpose though their functions are defined in the dataset. Then we have applied FunPred 3.0 to predict the functions of the test proteins. Predicted functions are hence matched with the original ones to compute precision, recall and F-score. While executing FunPred 3.0, it has been observed that 870 PPIs out of 13,528 protein-protein interactions (PPIs), that is, ∼6.4% of the overall MIPS dataset (Mewes et al., 2002) are unpredicted, that is, either unknown or missing. FunPred 3.0 has been also applied to predict the unannotated protein function and protein interaction (Mamoon, Sumathy & Gajendra, 2010), and also assigning the functional annotations for 767 PPIs out of 870 PPIs, representing circa ∼5.7% of the overall MIPS dataset. Similar instances have been also observed in the case of the subcellular location of proteins where 1,679 proteins out of 6,721 unique number of proteins are still unpredicted; the subcellular localization of these proteins are still unknown. The predicted functional annotations and subcellular localization of these unpredicted proteins and protein pairs, respectively result in relevant biological information, such as vital processes, diseases related mutations.

Methodology

In one of our two earlier works, Funpred-1 (Saha et al., 2014), the selection of 10% of test proteins of the top eight functional groups from the dataset was done randomly. Top eight functional groups were selected on the basis of maximum number of occurrences and interactions of proteins in them. While in another, FunPred 2 (Saha et al., 2017a), protein clusters are formed initially by the application of node and edge weight. Then 50% of proteins from each of the formed clusters are selected as test proteins. In both the cases, test proteins are chosen randomly. Since both these works are completely based on protein function prediction from PPIN, so network formed for each test protein is extensively large (up to level 2) enough to process which gradually enhances the overall computational overhead. Randomness is basically implemented to filter out the most essential proteins out of the entire PPIN and select them as the test protein. However, variation of test set (i.e., proteins beyond 10% in FunPred-1% or 50% in FunPred 2) as well as application of node and edge weight thresholds (in FunPred 2) might also play an important role in prediction accuracy level which has not been yet tested. It may be also considered as a major drawback and limitation of our previous two methodologies, which FunPred 3.0 has tried to overcome. FunPred 3.0 is an extended and advanced version of FunPred 2. The basic outlay of the both is the same, but the uniqueness of FunPred 3.0 can be defined in a three-way approach: Application of three levels of threshold for the formation of clusters and selecting all proteins from the clusters as test set to overcome the limitations of its predecessors.

Incorporation of feature selection.

Capable of prediction of functions of unpredicted protein pairs (incomplete and missing functional annotations) in MIPS dataset of Saccharomyces cerevisiae.

Capable of prediction of subcellular localization of proteins.

PPIN formed of proteins and their corresponding interactions may contain essential/non-essential proteins as well as reliable/unreliable edges. Proteins having maximum number of interconnected neighbors are considered as essential proteins while proteins having less number of interconnected neighbors are considered as non-essential ones. Presence of non-essential neighbors in the PPIN might affect the unknown protein function prediction level accuracy. Therefore, proper identification and elimination of non-essential proteins is needed to ensure the presence of maximum number of essential proteins in the PPIN. In the proposed work, detection of essential proteins is implemented by node weight. Node weight (Wang & Wu, 2013) basically assigns a weightage score to each node or protein based on its corresponding degree. High node weight determines essential while low node weight detects non-essential protein. Thus, non-essential proteins are discarded from PPIN along their corresponding edges. Even after this initial phase of PPIN refinement, there are still some unreliable edges present in the network. Since protein clusters are formed, so it is obvious that two nodes with an edge between them belong to the same cluster if they have high similarity. Edge between two nodes of high similarity is considered as reliable edge while that of low similarity is denoted as unreliable edge. In the proposed work, detection of reliable edges is executed by edge weight. Edge weight (Wang & Wu, 2013) also assigns a weightage score to each edge connected by two proteins in the terminals. Assignment of edge weight to an edge depends on the number of common neighbors between the two terminal proteins of the corresponding edge. More number of common neighbors signifies high similarity which in turn detects reliable edges. On the other hand, unreliable edges have low similarity since they have less number of common neighbors. Thus, unreliable edges are identified and pruned from the PPIN. Filtered out PPIN after these refinements, contains only essential proteins and reliable edges, which ultimately helps in enhancing prediction accuracy level.

In newly proposed algorithm, FunPred 3.0, first detects protein cluster and then selects all the proteins as test proteins from different predicted clusters. We have adopted the approach of forming protein cluster as mentioned in the work of Wang & Wu (2013). Protein clusters, thus formed, comprises of proteins belonging to any functional group. It results in accumulating larger number of functional groups as compared with only eight functional groups in our previous work (Saha et al., 2014). The novel computational method works in two stages: All the unique proteins are first clustered into M mutually exclusive clusters based on their node weight and edge weight in the overall PPIN. Node and edge weight have been used to ensure all the most essential nodes with higher reliability are present in the cluster and get selected as test proteins.

Functional annotations are then derived from the multi-level neighborhood of an unknown protein within each cluster.

More specifically, FunPred 3.0 is categorized into two sections: FunPred 3.0_Clust and FunPred 3.0_Pred.

FunPred 3.0_Clust uses the node weight and edge weight properties to rank and cluster all the proteins, creating M mutually exclusive protein clusters (Wang & Wu, 2013). The number of functional labels, associated with each interacting pair is large and in some cases annotations in each such cluster is unpredicted (incomplete or missing). This fact forces us to heuristically choose the node and edge weight threshold values, such that the unlabeled proteins are associated with larger protein clusters and have many neighborhood interactions (see Figs. 1 and 2). Three thresholds (high, medium and low) are set for each of node and edge weight using Eq. (1) (Zhang et al., 2016).

(1) Thk=α+k⋅σ⋅(1−11+σ2)

Figure 1 Filtering of PPIN.

Application of node weight and edge weight at three levels of threshold: High, Medium and Low in FunPred 3.0_Clust.

Figure 2 Cluster formations.

Formation of clusters from refined network after application of three levels of node and edge weight threshold in FunPred 3.0_Clust.

Where for node weight/edge weight, k ∈ {1, 2, 3} denotes three different thresholds, that is, low, medium and high, respectively. α is the mean of node weight/edge weight values of all proteins. σ is the standard deviation of node weight/edge weight values of all proteins. Proteins and edges having value less value than these node and edge weight thresholds get discarded and are considered as non-essential proteins and unreliable edges in the network, respectively.

The entire methodology has been described in Algorithm 1 as well as pictorially highlighted in Figs. 1 and 2. In Fig. 1, sample Table 1 (node weight table) is formed from the initial PPIN of yeast. Hence, node weight threshold is calculated using Eq. (1) at three levels: high, medium and low. These thresholds are applied on the initial PPIN to filter out three sub-networks, that is, sub-network1, sub-network2, sub-network3, respectively at high, medium and low node weight thresholds. Respective edge weight tables, that is, sample Table 2 (in Fig. 1), sample Table 3 (in Fig. 1), sample Table 4 (in Fig. 1) are formed from sub-network1, sub-network2, sub-network3, upon which high, medium and low edge weight thresholds (obtained using Eq. (1)) are applied to form pruned sub-network4, sub-network5 and sub-network3.

Table 1 Top-ranked selected physicochemical features (marked in blue and bold)-using four classifiers based on the maximum number of hits.

Physicochemical properties	Classifiers used
(Returns #5 top-ranked physicochemical properties/features)	
XGBoost	Random tree	Extra tree	Recursive feature elimination	#Hits	
Aromacity	✖	✓	✓	✖	2	
Gravy	✖	✓	✓	✓	3	
Instability index	✖	✓	✖	✖	1	
Isoelectric point	✓	✓	✓	✓	4	
Negatively charged particle	✓	✖	✖	✓	2	
Positively charged particle	✓	✖	✓	✓	3	
Extinction coefficient	✓	✖	✖	✖	1	
Aliphatic index	✓	✓	✓	✖	3	
Absorbance	✖	✖	✖	✓	1	
Ip/mol weight	✖	✖	✖	✖	0	

Table 2 Performance analyses of FunPred 3.0_Pred_SL.

Types of Proteins (based on Subcellular-localization)	Total no. of proteins in database	Total number of selected annotated proteins	Total number of selected essential test proteins	Prediction accuracy (Total no. of matched proteins)	Prediction accuracy (Total no. of unmatched proteins)	Failed to predict	
Nuclear proteins	1,771	1,609	162	112	32	18	
Cytoplasm proteins	1,757	1,566	191	109	51	31	
Interface proteins	2,246	2,176	70	37	23	10	

Table 3 Precision, recall and F-score obtained at three levels of node and edge weight threshold.

Threshold type	Node weight threshold	Edge weight threshold	Selected test proteins	Precision	Recall	F-score	
High	1.072	0.110	433	0.55	0.82	0.66	
Medium	1.068	0.109	433	0.55	0.82	0.66	
Low	1.064	0.107	520	0.54	0.82	0.65	

Table 4 Performance analyses of FunPred 3.0 with other protein function prediction methodologies.

Methods	Precision	Recall	F-score	
FunPred 3.0	0.55	0.82	0.66	
FunPred-2 (Saha et al., 2017a)	0.51	0.90	0.65	
FPred_Apriori (Prasad et al., 2017)	0.64	0.66	0.65	
FunPred 1.1 (Saha et al., 2014)	0.61	0.50	0.55	
FunPred 1.2 (Saha et al., 2014)	0.63	0.56	0.59	
Deep_GO (Kulmanov et al., 2018)	0.48	0.49	0.48	
Chi-square #1&2 (Hishigaki et al., 2001)	0.20	0.25	0.22	
Chi-square #1 (Hishigaki et al., 2001)	0.25	0.27	0.26	
Neighborhood counting #1&2 (Schwikowski, Uetz & Fields, 2000)	0.28	0.41	0.33	
Neighborhood counting #1 (Schwikowski, Uetz & Fields, 2000)	0.26	0.45	0.33	
Fs-weight #1&2 (Chua, Sung & Wong, 2006)	0.36	0.43	0.39	
Fs-weight #1 (Chua, Sung & Wong, 2006)	0.33	0.42	0.37	
Nrc (Moosavi, Rahgozar & Rahimi, 2013)	0.37	0.43	0.40	
Zhang (Zhang et al., 2009)	0.20	0.19	0.19	
DCS (Peng et al., 2014)	0.36	0.37	0.36	
DSCP (Peng et al., 2014)	0.39	0.40	0.39	
PON (Liang et al., 2013)	0.15	0.14	0.14	

In Fig. 2, sample Table 5 (node weight table formed from sample Table 1 under high threshold in Fig. 1) has been generated from refined sub-network4 which afterward is sorted in descending order to form sample Table 6 (in Fig. 2). From sample Table 6 (in Fig. 2), first protein having the highest node weight is selected as seed of the initial cluster. Then corresponding level 1 neighbors of the seed are included in the cluster provided its inclusion in the cluster does not let the edge weight to fall below the high threshold (verified using sample Table 2 in Fig. 1). The entire initial cluster contents are discarded from sample Table 6 (in Fig. 2) and the next cluster is formed with the next selected seed. This process continues until all the proteins in sample Table 6 (in Fig. 2) get clustered. Thus M mutually exclusive protein clusters are formed where 1 =< M <= N4 (N4 is the total number of proteins present in sample Table 6 in Fig. 2). The entire procedure is repeated for refined sub-network5 and refined sub-network6 until M mutually exclusive protein clusters is formed for each of them.

Table 5 Predicted samples of unpredicted protein pair interactions/functions (“missing” protein-pair-interactions/functions) in the MIPS dataset.

Interacting protein pairs	Predicted interactions	Predicted functions	
Protein#1	Protein#2	Interaction#1	Interaction#2	Function#1	Function#2	
YAL014c	YAL030w	Two hybrid	Coimmunoprecipitation	–	–	
YAL014c	YMR197c	Two hybrid	Coimmunoprecipitation	–	–	
YLR459w	YDR434w	Unable to Predict	–	–	–	
YDR167w	YBR081c	Two hybrid	–	–	–	
YGL173c	YML085c	Synthetic lethal	–	–	–	
YGL190c	YKL048c	Synthetic lethal	Two hybrid	Cell polarity	–	
YMR167w	YNL082w	Coimmunoprecipitation	Copurification	DNA repair	–	
YDR027c	YJR060w	Affinity chromatography, affinity-tag GST	Two hybrid	–	–	
YGR082w	YNL131w	Crosslinking	Coimmunoprecipitation	–	–	
YJR066w	YHR186c	Synthetic lethal	–	–	Lipid metabolism	
YDR363w-a	YER008c	Synthetic lethal	–	Vesicular transport	–	
YDR309c	YLR319c	Synthetic lethal	Cell structure	–	–	
YLR336c	YPL268w	Unable to predict	–	–	–	
YKR099w	YDL106c	Unable to predict	–	–	–	

Table 6 Predicted samples of unpredicted protein pair interactions/functions (“unknown” protein-pair-interactions/functions) in the MIPS dataset.

Interacting protein pairs	Predicted interactions	Predicted functions	
Protein#1	Protein#2	Interaction#1	Interaction#2	Function#1	Function#2	
YLR418c	YIL040w	Two hybrid	–	Pol II Transcription	–	
YOR326w	YNL120c	Mitosis	–	Cell polarity	Cell cycle control	
YJR057w	YDR438w	Unable to Predict	–	–	–	
YFL037w	YMR299c	Cell structure	–	RNA processing	DNA repair	
YHR129c	YGL124c	Mitosis	Two hybrid	–	–	
YGR078c	YAL011w	Synthetic lethal	Two hybrid	–	–	
YNL153c	YDR149c	Two hybrid	–	Pol II transcription	–	
YMR307w	YMR317w	Two hybrid	–	Carbohydrate metabolism	–	
YLR039c	YIL039w	Vesicular transport	Two hybrid	–	–	
YMR307w	YHR004c	Two hybrid	–	Carbohydrate metabolism	–	
YDL003w	YGL250w	Two hybrid	–	Energy generation	–	
YNL271c	YGR228w	Meiosis	–	Cell polarity	Protein modification	
YML094w	YBR108w	Unable to predict	–	–	–	
YEL003w	YDR334w	Unable to predict	–	–	–	

All the proteins belonging to M mutually exclusive protein clusters (under three levels of thresholds: high, medium and low separately), obtained from FunPred 3.0_Clust are considered as essential proteins and hence they are included in the test set of our proposed methodology. In the second step, FunPred 3.0_Pred predicts labels these selected unlabeled (or test) proteins using neighborhood properties and physicochemical properties of amino acids (see Algorithm 2 and Fig. 3). In Fig. 3, for each test protein (say P1 belonging to refined sub-network4 in Fig. 1) its level 1 neighborhood graph is formed including those proteins which are present in its corresponding node weight table: sample Table 1 under high threshold in Fig. 1. Then protein clusters are formed at level 1 (considering each level 1 protein as the seed of the cluster) in a similar way as formed in FunPred 3.0_Clust. It should be noted here that the number of clusters formed here is equivalent to the number of proteins in level 1. Distance between the mean of the physico-chemical features of each protein cluster as well as test protein is computed and the test belongs to the cluster having the least distance. All the functions of the selected cluster are allocated to the test protein.

Figure 3 FunPred 3.0_Pred.

Working Model of FunPred 3.0_Pred. A: Selected test protein B: Formation of PPIN of test protein C: Formation of clusters D: Computation of distance of the test protein from each of the formed cluster E: Allocation of test protein to the selected cluster having minimum distance along with all it’s functions.

The relevant level 1 neighbors of the test proteins are chosen to form their individual neighborhood graph. In finding the level 1 neighbors or forming their individual neighborhood graph, relevance is measured in terms of edge weight properties. Next, PCP score is computed for the neighborhood graph of each test protein. Six different high-ranked physico-chemical features: aliphatic index (Singh, Wadhwa & Kaur, 2008), gravy (Kyte & Doolittle, 1982), aromacity (Lobry & Gautier, 1994), number of negatively charged residues (Singh, Wadhwa & Kaur, 2008), number of positively charged residues (Singh, Wadhwa & Kaur, 2008), isoelectric point (Bjellqvist et al., 1994) are used to reckon this physico-chemical property (PCP) based score. These high-ranked features are selected from 10 divergent physico-chemical features (see Supplementary) by the enactment of four distinct classifiers: XGBoost classifier (Chen & Guestrin, 2016; Pedregosa et al., 2011), Random Forest classifier (Breiman, 2001; Pedregosa et al., 2011), Extra Tree classifier (Geurts, Ernst & Wehenkel, 2006; Pedregosa et al., 2011) and Recursive feature elimination (RFE) classifier (Pedregosa et al., 2011). High-ranked five among 10 features have been picked at first by each classifier. Then from these picked features, frequency of maximum occurrences for each individual feature has been noted from which endmost six features get selected (see Table 1). Finally, each test protein is assigned to a functional group of the neighborhood graph, based on the nearest neighborhood approach on the basis of mean PCP score. FunPred 3.0_Clust (see Algorithm 1) and FunPred 3.0_Pred (see Algorithm 2) describes the methodology of unknown protein selection and function prediction, respectively.

Algorithm 1 FunPred 3.0_Clust: (For formation of protein clusters which consist of essential proteins and reliable edges).

Input: Undirected PPIN G.	
Output: Protein clusters at three levels of threshold: high, medium and low	
 Begin	
   //computation of node weight of G	
   for all nodes in G	
    compute node weight	
   //computation of node weight threshold	
   compute node weight threshold at three levels: high, medium and low using equation 1	
   //Elimination of non-essential proteins based on node weight threshold	
   for each level of threshold	
    for all nodes in G	
      if node weight does not exceed threshold	
       remove corresponding node.	
   //Formation of refined sub-networks G′high, G′medium and G′low from G	
   G′high, G′medium and G′low consisting of only essential proteins (high node weight) are formed	
   //computation of edge weight of G′	
   for all edges in G′high, G′medium and G′low	
    compute edge weight	
   //computation of edge weight threshold	
   compute edge weight threshold at three levels: high, medium and low using equation 1	
   //Elimination of unreliable edges based on edge weight threshold	
   for all edges in G′high	
    if edge weight does not exceed high level of edge threshold	
      remove corresponding edge.	
   repeat the same for low, medium level of threshold and G′medium and G′low respectively.	
    //Formation of refined sub-networks G″high,high, G″medium,medium and G″low,low from	
   G′high, G′medium and G′low at high node and edge weight threshold, medium node and edge weight threshold, low node and edge weight threshold respectively.	
   form G″high,high, G″medium,medium and G″low,low consisting of only reliable edges (high edge weight)	
   //Formation of clusters at three levels of thresholds	
   for all proteins in G″high,high	
    form node weight table	
   sort the node weight table based on the node weights	
   select the first protein P in the node weight table as the seed of initial cluster CMhigh	
   i.e. CMhigh = {P} where 1=<M<=W (W is the total no. of nodes in node weight table)	
   neighbors of P are added to CMhigh provided its inclusion does not cause edge weight to fall below high edge weight threshold value i.e. CMhigh={P}∪NP1	
   update the node edge table by eliminating all the proteins present in CMhigh and continue with the next seed to form clusters in the same way mentioned above till all the proteins in the node weight table belongs to a cluster.	
   repeat the same procedure for G″medium,medium and G″low,low.	
 End	

Algorithm 2 FunPred 3.0_Pred: (Protein function prediction of test proteins).

Input: Set of un-annotated proteins in CMhigh, CMmedium, CMlow selected by FunPred 3.0_Clust	
Output: Functional group of un-annotated proteins	
 Begin	
   // Formation of clusters at level −1 of un-annotated protein	
    for each protein P in CMhigh	
      for each level −1 neighbor N of P present in G′high	
       add N as the seed of the cluster Ki i.e. Ki = {N}.	
       //where 1=<i<=g, g is the total number of level – 1 neighbors of P	
       add immediate neighbors of N i.e. INN in the cluster Ki provided such inclusion does not cause the edge weight to fall below the high edge weight value of threshold as computed in . FunPred 3.0_Clust i.e. Ki = {N} ∪ INN.	
   //Feature selection	
    compute Physico-Chemical features of each protein from the amino acid sequence of each protein and execute the selected classifiers to select the most essential features.	
    //Here six features get selected as the essential ones from the initial list of ten features.	
    //Computation of PCPscore	
    for each protein P in CMhigh	
    compute its mean PCPscore of six selected Physico-Chemical features	
    for each formed cluster Ki of protein P	
      compute its mean PCPscore of six selected Physico-Chemical features	
   //Assigning of Functional Groups to the proteins in CMhigh	
    for each protein P in CMhigh	
    for all clusters Ki of protein P	
      obtain the difference of PCPscore of P and clusters Ki	
    functional groups of cluster Ki are assigned to protein P having least difference.	
    Repeat all the above steps for annotation of protein functions in CMmedium and CMlow	
 End.	

It needs to be highlighted here that both FunPred 3.0_Clust and FunPred 3.0_Pred have been executed at three levels: (1) high node and edge weight threshold, (2) medium node and edge weight threshold, (3) low node and edge weight threshold. So, we have tested FunPred 3.0 at each of the three levels to assess its performance impact.

Besides predicting protein function, it has been observed that the protein subcellular localization is yet another important aspect which needs to be considered since it helps in better understanding of protein function. So, subcellular localization dataset of yeast has been obtained from UniProt database (Apweiler et al., 2004). On careful observation it has been noted that there are 6,721 unique number of proteins out of which localization of some proteins are still unpredicted. It is similar as that of 6.4% of the overall MIPS dataset (Mewes et al., 2002) which are unpredicted, that is, either unknown or missing. So FunPred 3.0 is also implemented to predict these unpredicted protein subcellular localization.

In FunPred 3.0, the subcellular localization dataset of yeast is centrally categorized under three major sections: proteins residing in nucleus (termed as nuclear proteins), proteins residing in cytoplasm (termed as cytoplasm proteins) and proteins residing in other regions (termed as interface proteins). Besides these, there is also another section termed as unpredicted localization proteins, consisting of those whose localization are not yet predicted (see Figs. 4–6). Therefore, before dealing with the unpredicted localization proteins, the predictive accuracy of FunPred 3.0 needs to be assessed just in a similar way earlier as that of protein function prediction. The same test set of essential proteins (as generated by FunPred 3.0_Clust) of yeast (MIPS) is also considered here (localization of which are known but considered to be unknown for experimental purpose). Selected test and candidate proteins in the PPIN of nuclear, cytoplasm and interface proteins are highlighted in Figs. 7–9, respectively. Now for each test protein, its corresponding level 1 neighborhood graph is formed. In a PPIN, a protein almost shares similar properties as that of its neighborhood proteins. The same is also applicable to the test protein but all the properties or functions of the neighborhood cannot be transmitted to it. Therefore, proper assessment needs to be implemented in the neighborhood of the test protein. For this purpose, FunPred 3.0_Pred_SL (SL stands for Subcellular Localization) is applied. It first assigns respective subcellular localization information to the neighborhood of the test protein using UniProt database. Hence, it counts the frequency of occurrence of each subcellular location (nucleus, cytoplasm or any other region). The subcellular location having the highest frequency of occurrence among the neighborhood is allocated to the test protein. The test protein becomes nuclear or cytoplasm or interface proteins according to the allocated subcellular location. Then, the allocated subcellular localization is checked from the Uniprot database. The overall result which is achieved by the application of FunPred 3.0_Pred_SL, has been highlighted in Table 2. An overall accuracy of 69.1%, 57% and 53% is reckoned for nuclear, cytoplasm and interface proteins, respectively. It is observed from Table 2, that our method fails to predict the subcellular localization of few proteins among all the three categories. This is because our method mainly predicts subcellular localization using PPIN neighborhood based approaches and if there is practically no information or significantly less amount of interactive information in the PPIN for a particular test protein then our method fails. This can be considered as one of our limitations which can be redressed in future works by incorporating protein sequence.

Figure 4 Categorization of proteins based on subcellular localization.

PPIN of yeast (Saccharomyces cerevisiae): cytoplasm proteins (red), nuclear proteins (green), interface proteins (blue), unpredicted localization proteins (orange).

Figure 5 Network view of PPIN of yeast.

Sequential formation of cytoplasm proteins (red), nuclear proteins (green), interface proteins (blue), unpredicted localization proteins (orange) in PPIN of yeast.

Figure 6 Disintegrated network views of PPIN of yeast.

Separate PPIN’s of cytoplasm proteins (red), nuclear proteins (green), interface proteins (blue), unpredicted localization proteins (orange) and their interactions.

Figure 7 Nuclear PPIN of yeast.

Candidate (green) and test (yellow) proteins in nuclear PPIN (green and yellow) of yeast (violet: other nodes in the network).

Figure 8 Cytoplasm PPIN of yeast.

Candidate (red) and test (yellow) proteins in cytoplasm PPIN (red and yellow) of yeast (violet: other nodes in the network).

Figure 9 Interface PPIN of yeast.

Candidate (blue) and test (yellow) proteins in Interface PPIN (blue and yellow) of yeast (violet: other nodes in the network).

Subcellular localization dataset of yeast contains 1,679 number of unpredicted localization proteins out of which our method predicts the localization of 638 proteins successfully. Localization information for the remaining 1,041 proteins cannot be predicted because of the absence of PPIN interaction in MIPS dataset as discussed earlier. This extra added layer of biological information about subcellular localization of proteins along with the protein function prediction boost up our methodology FunPred 3.0 to the next higher level.

Results

Initially, PPIN of yeast consists of 4,554 unique proteins in 13,528 protein-protein interactions (PPINs) after the elimination of the self-replicating and the self-interacting protein pairs. After the network refinement through the execution of node and edge weight threshold, non-essential proteins along with unreliable edges get eliminated and the initial PPIN gets reduced to almost 3,174 unique proteins and 6,936 PPINs (approx.) considering three levels of thresholds from which FunPred 3.0_Clust form protein clusters to generate test set of proteins.

During the result analysis it is observed that proteins belonging to random functional groups like lipid metabolism, DNA Repair etc., get selected as test proteins. In FunPred 3.0_Pred, all proteins from each protein cluster formed from FunPred 3.0_Clust are considered as test proteins. The overall initial PPIN of yeast is highlighted in Fig. 10 while in Fig. 11, 433 test proteins (most essential ones among the refined PPIN of yeast consisting of 3,174 unique proteins and 6,936 PPINs) selected by FunPred 3.0_Clust at high node and edge weight threshold values, are highlighted in yellow circle (shape) in the initial PPIN of yeast consisting of 4,554 unique proteins in 13,528 protein-protein interactions (PPINs). It should be noted in Fig. 11, that all the selected test proteins belongs to the most densely connected region of PPIN which establishes the fact that these are indeed most strongly connected essential proteins. The performance of FunPred 3.0 is evaluated using standard performance measures, such as precision (P), recall (R) and F-score (F) values, which are calculated using the following equations:(2) P=TPTP+FP R = TPTP+FN F = 2 ∗ (P ∗ R)P+R

where TP, FP, FN represent True Positive, False Positive and False Negative, respectively.

Figure 10 Network view.

PPI network of Yeast (Saccharomyces cerevisiae).

Figure 11 Selected candidate and test proteins.

PPIN of annotated (red circle) and test/unannotated proteins (yellow circle) of the yeast network (Saccharomyces cerevisiae).

The performance of FunPred 3.0 has been analyzed under different levels of thresholds of node and edge weights as highlighted in Table 3. It should be noted here that under high and medium thresholds, the same precision, recall and F-score have been retrieved since number of selected test proteins are equivalent in both the cases. The result analysis as depicted in Table 3 shows that there are not many significant changes in the result varying the thresholds except for a slight fall in precision and F-score under low threshold as compared to the others. The fall is due to the relaxation in the node and edge weight thresholds resulting in incorporation of less essential proteins in the test set. So, the high threshold ensures inclusion of the most essential proteins. So the precision, recall and F-score of FunPred 3.0 are reckoned as 0.55, 0.82 and 0.66, respectively under high threshold. High recall and low precision emerges out as a major characteristic of FunPred 3.0 when compared to the other methodologies except FunPred-2. It highlights the fact that most of the admissible results are successfully generated by FunPred 3.0. Table 4 shows a detailed performance comparison of other methodologies along with our proposed systems (like FunPred 1.1, FunPred 1.2 (Saha et al., 2014)), the neighborhood counting method (Schwikowski, Uetz & Fields, 2000), the Chi-square method (Hishigaki et al., 2001), a recent version of the neighbor relativity coefficient (NRC) (Moosavi, Rahgozar & Rahimi, 2013), FPred_Apriori (Prasad et al., 2017), Zhang methodology (Zhang et al., 2009), domain combination similarity (DCS) (Peng et al., 2014), domain combination similarity in context of protein complexes (DSCP) (Peng et al., 2014), protein overlap network (PON) (Liang et al., 2013), Deep_GO (Kulmanov et al., 2018) and the FS-weight based method (Chua, Sung & Wong, 2006)). All these data are collected from their respective works which are executed on the same organism, that is, yeast. The results of Deep_GO (Kulmanov et al., 2018) are computed manually for the yeast dataset, the code of which is available at https://github.com/SovanSaha/FunPred-3.0.git. From Table 4, it can be also highlighted that our method, FunPred 3.0, yields relatively higher F-score values than the others including its earlier version FunPred-2.

Table 4 also discloses the fact that NRC method has overpowered the rest except FunPred 1.1 (Saha et al., 2014), FunPred 1.2 (Saha et al., 2014), FPred_Apriori (Prasad et al., 2017) and FunPred 3.0. The reason behind this is observed as follows: Both version of FunPred 1 has incorporated two levels (i.e., level 1 and level 2) of PPIN as well as lot of essential neighborhood properties like neighborhood ratio, protein path connectivity and relative functional similarity (includes both ancestor and descendant information of a specific protein) have been utilized to assess the reliability of each node (protein) along with its associated edges (protein interaction) during the unannotated protein function prediction. FPred_Apriori (Prasad et al., 2017) executes both closeness centrality and edge clustering coefficient to make its predictive approach more effective than the others. Last but not least, FunPred 3.0 combines PCP of each protein along with neighborhood analysis (like node weight, edge weight etc.) for predicting protein function which ultimately promotes it to the next higher level in the terms of performance analysis when compared to the others.

Though the neighborhood counting method is simple in nature yet the performance measure of it has descended considerably in comparison to NRC (Moosavi, Rahgozar & Rahimi, 2013), FS-weight #1 (directly connected proteins) and FS-weight #1 and #2 (directly and indirectly connected proteins) despite of its simplicity (Chua, Sung & Wong, 2006). This is because no differentiation has been observed between the direct and indirect neighborhood connection. Beside most of the methods included in Table 4 like NRC (Moosavi, Rahgozar & Rahimi, 2013), Chi square #1&2 (Hishigaki et al., 2001), Chi square #1 (Hishigaki et al., 2001), Neighborhood counting #1&2 (Schwikowski, Uetz & Fields, 2000), Neighborhood counting #1 (Schwikowski, Uetz & Fields, 2000) etc., are not utilized for the refinement of the PIN by pruning unreliable proteins or edges which in turn increases false positives in their prediction accuracy level. In FPred_Apriori (Prasad et al., 2017), a bottom-up predictor of existing Apriori algorithm has been utilized for protein function prediction by exploiting two most important neighborhood properties: closeness centrality and edge clustering coefficient of protein interaction network. Though the method is unique in the fact that the functions of the leaf nodes in the interaction network have been back propagated and thus labeled up to the root node (test protein) but yet it fails to generate high Recall and F-score than FunPred 3.0. But it returns substantially high precision values than the others as well as all our methods. DCS (Peng et al., 2014), DSCP (Peng et al., 2014), PON (Liang et al., 2013), Deep_GO (Kulmanov et al., 2018) and Zhang methodology (Zhang et al., 2009) are well developed methods for protein function prediction incorporating domain specific as well as neighborhood based properties but they fail to compete with all our methodologies due to the lack of important feature selection methodologies of physicochemical properties and proper assessment of nodes and edges involved in test set protein function prediction through node and edge weights.

During experimental evaluation, the validation set is prepared with 4,554 labeled S. cerevisiae proteins, collected from the MIPS dataset. Using FunPred 3.0_Clust, we identify M mutually exclusive protein clusters (Wang & Wu, 2013). Experimental variations with K = 1,2,3 are included in Table 2. Using an optimal choice of K = 3, we identify 433 test targets for the validation set. Now, the functional labels of these test proteins are assigned using FunPred 3.0_Pred. The precision, recall, F-scores of our method over the test targets of the validation set is obtained as 0.55, 0.82 and 0.66, respectively.

Discussion

Our results (characterized by the precision, recall and F-score) and comparison with the other protein functional group prediction models show the superiority of our approach. The FunPred 3.0 software has better performance than any existing function prediction in silico method. The network structure may be pruned based on the edge weight and along with it use of physico-chemical properties lead to improved and faster functional prediction in complex and diverse protein–protein interaction networks. We would like to estimate the effectiveness of our in silico method for other organisms, such as in human protein–protein interactions with even more complex network architectures.

The initial results motivate us to predict the subcellular localization and unpredicted protein pair functions (missing/unknown functions) for 870 PPIs extracted from MIPS dataset. A protein can perform multiple functions in isolation. It may also perform some specific functions while interacting with one protein while perform some other specific functions while reacting with other proteins. But considering the fact that a protein often shares similar functions with proteins that interact with it (Chakicherla et al., 2009; Chatterjee et al., 2012; Shatsky et al., 2016), each protein pair is disintegrated in two constituent proteins and functions of each protein is predicted using FunPred 3.0. For an unknown protein pair P1P2, we predict the functions as an intersection of FunPred 3.0_Pred(P1) ∩ FunPred 3.0_Pred(P2). The results of all the predicted annotations for the MIPS dataset are available at https://github.com/SovanSaha/FunPred-3.0.git. Examples of the prediction of protein function and interactions for unpredicted pairs (both unknown and missing protein pair) have been shown in Table 5 and Table 6, respectively.

Summarizing, 767 unpredicted protein pair functions (511 unknown protein pair functions and 256 missing protein pair functions) in the MIPS dataset could be predicted using our FunPred 3.0 algorithm. Our approach failed to predict 103 unpredicted protein pairs since they have less number of acceptable neighbors. Simultaneously our methodology also performs very well in predicting subcellular localization of proteins as discussed earlier in the methodology section earlier. All the datasets and supplementary files are also freely available at https://github.com/SovanSaha/FunPred-3.0.git.

Conclusion

FunPred 3.0 proved to be an improved and advanced version of our previous methodology FunPred-2. The enhanced performance of FunPred 3.0 is due to the use of node weight, edge weight and physicochemical properties of proteins in the prediction pathway of test set of proteins. It should be highlighted here that FunPred 3.0 incorporates the most essential features classified through four classifiers: XGBoost, Random Forest, Extra Tree and RFE. RFE which indeed plays an important role in improving the performance of the proposed methodology. This method does not consider dynamic PPIN and integration of other multiple types of data like domain (Chatterjee et al., 2011a, 2011b) etc., but topological analysis, association between function and protein have been proven to be significant for this research. The use of FunPred 3.0 to detect the subcellular localization of proteins as well as the function of unpredicted protein pair functions (unknown and missing pairs of proteins) in the MIPS database add an extra dimension to this work. Incorporation of other protein-related features and their integration and the use of the other benchmark datasets for different organisms may give a proper insight for prediction. Beside this unannotated protein function prediction, the methodology behind the FunPred 3.0 algorithm can be also used in disease-specific datasets (Saha et al., 2017b), which also may be a future direction. In a nutshell, the work presented here proposes the statistical learning evaluation of various features for prediction of protein functions in the complex yeast PPIN with reasonable accuracy. The dataset used in this study and the complete source codes of the FunPred 3.0 software package are available in the public domain (https://github.com/SovanSaha/FunPred-3.0.git) for non-commercial research.

Supplemental Information

Supplemental Information 1 Basic terminologies and information about FunPred 3.0.

Information about Protein-Protein interaction network, Protein pair function prediction, MIPS database and physico-chemical properties.

Click here for additional data file.

Additional Information and Declarations

Competing Interests

Author Contributions

Data Availability

The authors declare that they have no competing interests.

Sovan Saha conceived and designed the experiments, performed the experiments, analyzed the data, contributed reagents/materials/analysis tools, prepared figures and/or tables, approved the final draft.

Piyali Chatterjee conceived and designed the experiments, performed the experiments, analyzed the data, contributed reagents/materials/analysis tools, prepared figures and/or tables, approved the final draft.

Subhadip Basu conceived and designed the experiments, analyzed the data, prepared figures and/or tables, authored or reviewed drafts of the paper, approved the final draft.

Mita Nasipuri conceived and designed the experiments, analyzed the data, authored or reviewed drafts of the paper, approved the final draft.

Dariusz Plewczynski conceived and designed the experiments, analyzed the data, authored or reviewed drafts of the paper, approved the final draft.

The following information was supplied regarding data availability:

The code is available as a Supplemental File together with training datasets and the results. All resources and materials are additionally available at: https://github.com/SovanSaha/FunPred-3.0.git

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
