# Peer review of "FunPred 3.0: improved protein function prediction using protein interaction network"

_PeerJ, doi:10.7717/peerj.6830_

## Round 0.1 · original submission · Major Revisions

The manuscript has received some substantial criticisms from the reviewers and even a recommendation to reject. However I believe it could be revised to become Acceptable, therefore I invite you to take on these comments and consider submitting a major revision of the manuscript.

Reviewer 1 ·

Basic reporting

The manuscript introduces a new version, 3.0, of an existing program, FunPred (by the same authors) for protein function prediction.

Experimental design

The methods are very poorly described. I'll give some examples:

L92-93: "While in another, FunPred 2 (Saha et al. 2017a), fifty percent of
93 proteins are selected as test proteins from each formed cluster."
What cluster is this referring to? No cluster has been mentioned before.

L97: "strongly connected essential proteins"
"Strongly connected" usually refers to graphs. What is a strongly connected protein?

L105-106: "Application of three levels of threshold for the formation of clusters and selecting all proteins from the clusters as test set to overcome the limitations of its predecessors."
This is the first difference between FunPred 3.0 and FunPred 2.0. The choice of the test set is an evaluation difference, not a design difference.

Fig.1: "M clusters are formed, the proteins of which are selected as test proteins"
This seems to imply that ALL proteins in the clusters are chose as test proteins, which does not make sense.

Fig.3: the diagram is highly redundant, without clarifying what the program does.

I cannot say I understand what the new algorithm does.

Validity of the findings

The results are not impressive at all.
First, Table 2 shows almost no difference between the precision and recall obtained for the three different thresholds.
Second, and most important, Table 3 shows negligible F-score improvement w.r.t. two existing programs.

Additional comments

The new algorithm has to be clearly described. Also, since there is negligible improvement of the state of the art, the introduction of a new algorithm has to be justified.

Reviewer 2 ·

Basic reporting

The introduction has a good discussion of the context of the problem and similar work that this manuscript builds on. The language needs minor editing for grammar at a few places.

The methods, however, are not described in insufficient detail for certain points of the analysis. For example:
1. The definitions of the node and edge weights are described in the cited paper and the supplementary methods but the main text doesn’t say what these weights are or where to find these.
2. Line 125 says “… creating M mutually exclusive protein clusters…”, but the value of M (number of clusters) is never mentioned. In addition, in the algorithm in Figure 1, it is not clear whether the algorithm proceeds until all the proteins have been clustered, and the number of clusters simply depend on the chosen thresholds, or whether M can be controlled.
3. The selection of the test set proteins is also not clear. Line 105 says “ … and selecting all proteins from the clusters as test set …”, which either implies that all the proteins are part of clusters and are part of the test set, or only those proteins that are part of the clusters can be part of the test set. Figure 6 implies a conventional selection of a small set of the proteins as the test set. All of these statements and the Figure are not consistent which each other.
4. It is not clear what is meant by “ … prediction of unknown/missing protein pair functions in MIPS dataset …”. Does it mean prediction of missing protein pair interactions? If it is prediction of protein function, why are these pairs of proteins supposed to have the same function. Conventionally, the functions of single proteins are predicted. What if a protein is part of multiple pairs and each has a different function?
5. Table 4 doesn’t clear the confusion. Pairs of proteins are predicted, but some of the predictions are functions or biological processes the proteins might be involved in, such as cell polarity, vesicular transport, DNA repair etc., while others such as two hybrid and synthetic lethality are interactions. Prediction of protein function and the prediction of protein interactions are very different problems and should be conceptually separated, even if the underlying algorithm is similar for this method. Table 5 again mixes these two modalities.
6. The protein functions of the annotated proteins, and consequently, the predicted functions should be clearly defined prominently where the interaction dataset is defined. Is this the Gene Ontology BP or MF set? Are these functions for which the best features are found, as described in lines 148-159?

Experimental design

The basic idea of clustering the proteins into clusters, and then predicting protein functions using the cluster that is “closest” to a protein in terms of some network distance and/or physiochemical properties is valid. However, it is hard to evaluate the method in detail when the method Is inadequately described (as mentioned in the Basic Reporting review section). At the very least, the following points should be cleared:
1. The selection of the number of protein clusters.
2. The separation of protein function prediction and protein interaction prediction.
3. If protein function is done for pairs of proteins, this is unconventional, and it needs an explanation for its logic and a discussion of why this is a good idea.

At this stage, the only way to get answers is to dig deep into the provided code. But the manuscript should provide at least the conceptual framework.

Validity of the findings

It is hard to evaluate the validity of the findings while questions remain about the experimental design. As mentioned earlier Tables 4 and 5 mix two different modalities of prediction.

---

## Round 0.2 · Major Revisions

The manuscript needs further substantial revision. It is close to rejection. However I believe it could be revised. Since large work done already this manuscript will find its readers. It either needs substantial revision or resubmission as a new manuscript.

Reviewer 1 ·

Basic reporting

N/A (revised version)

Experimental design

N/A (revised version)

Validity of the findings

N/A (revised version)

Additional comments

I appreciate the authors' efforts to improve the quality of the presentation. However, the authors agree with me on the most important aspect: the new program presents negligible improvement over the previous version:

"Since FunPred 2 is modified to FunPred 3.0 and the basic working mechanism of the both remains same, negligible improvement in the F-Score has been observed in Table 3 though major improvement has been observed when compared to the basic version FunPred 1."

Since FunPred2 has been already published in all details, FunPred3 is in fact an incremental improvement, more appropriate to be implemented only in the github webpage of the software.

Reviewer 2 ·

Basic reporting

The updated manuscript has added important information to attempt to answer some of the questions raised in the previous review. However, there are still many questions regarding the methods, and describing the methods in sufficient detail is essential for a potential reader to gain useful knowledge from the article.
In the following, I will touch on some of these points.
Major points:
1. Line 76 states that a certain number of interactions are either unknown or missing. What does this mean? If a PPI interaction is not listed in MIPS, how does one conclude whether a) this pair of proteins do not interact, or b) this is a missing interaction? What are unknown interactions? Bioinformatics databases list the experimental modality and source for each interaction. So what are these unknown interactions? Does this have something to do with unknown functions of proteins? What is the criteria for calling a pair of proteins to have unknown function? When one of the proteins has unknown function or both of them?
2. It is not explicitly mentioned how the True Positives, False Positives etc. are calculated in line 267. Are these for the presence/absence of a protein-protein interaction or for the functional classification of proteins? If it is for the protein function (as can be assumed the title and text), what/how many are these functional categories considered? The article mentions both Gene Ontology and MIPS Functional Catalogue. Which of these is used? Both GO and the MIPS functional catalogue are hierarchical. How is the function prediction of a wider/narrower category handled? For example, a protein if a protein has been annotated to MIPS function 11.06.03.01 (mRNA editing)is it counted as a true positive if the algorithm predicts its annotation to be 11 (transcription)? Or are these categories taken at a particular level, such as the GO slim categories?
3. The language/sentence structure should also be improved, but this can be done by a proof-reader or language editor.

Minor points:
1. Please consider using the more familiar terminology of a test set/test instance, cross-validation set or a hold out set rather than a target protein.

Experimental design

Broadly, the methodology seems within reason, but the experimental design can only be evaluated completely after the methods have been unambiguously described.

Validity of the findings

Dependent on questions raised earlier.

---

## Round 0.3 · Minor Revisions

Please check remaining technical remarks. I'd recommend add recent literature on associative networks - that also serve for protein functional prediction. Important note - please, check again English presentation. I'd recommend make sentences shorter, more readable.

Reviewer 2 ·

Basic reporting

I thank the authors for their detailed answers to the questions raised in the previous review. Since the mixing of the experimental modality and function, and prediction of protein functions of pairs of proteins rather than individual proteins is unusual, it will be helpful if the explanation written for the review is made available to the readers, perhaps as a supplementary methods file. At this point, I feel that the method has been adequately described that a discerning reader will be able to understand the results.

I just have a few minor suggestions for the authors.

Line 330 should read “Last but not least”, rather than “Last but not the least”. While discussing the other methods in lines 334 to 351, it’s better to include the citation for each method, even though they’ve been included earlier in lines 313 to 319.

Are the performance measures of the other methods listed in Table 4 taken from the different papers, or are they computed by the authors on the same test set chosen for this work? Please describe this somewhere. If they were computed on the same dataset, it would be nice to include the code for this in the repository. The captions for Tables 5 and 6 can be edited to clearly mention where the interacting pairs are being predicted, and where the function is being predicted while the pair is already listed in MIPS.

Experimental design

no comment

Validity of the findings

I am not sure whether this algorithm as reported will be used for practical function/pair prediction, but the authors have reported a valid method and described it for other researchers. Some of the ideas presented here may be used by other researchers in future work.

---

## Round 0.4 · accepted · Accept

After several review rounds this work was significantly updated. I think it could be published now.

# Reviewer 2 ·

Basic reporting

I think the language/readability of the text is barely satisfactory, but detailed language editing is outside the scope of this peer review.

Experimental design

no comment

Validity of the findings

The results in Table 4 are copied from the different publications, and this should also be indicated in the table caption. As such, the different studies probably use different test and training set splits than the authors. Therefore, a direct comparison will not necessarily support the superiority of this method. The corresponding statement in the first sentence of the discussion section should be removed/modified.

The columns 5 and 6 in Table 2 should probably not be called Prediction Accuracy. Accuracy is usually understood as a fraction or percentage of the correct predictions.

Tables 5 and 6 caption should also mention whether these are all the unknown/missing interactions, a random sampling, or selected based on some scoring.